# Reward Scale Robustness for Proximal Policy Optimization via DreamerV3 Tricks

**Ryan Sullivan**
rsulli@umd.edu
University of Maryland
College Park, MD, USA

**Akarsh Kumar**
akarshkumar0101@gmail.com
Massachusetts Institute of Technology
Cambridge, MA, USA

**Shengyi Huang**
costa.huang@outlook.com
Drexel University
Philadelphia, PA, USA

**John P. Dickerson**
johnd@umd.edu
University of Maryland
College Park, MD, USA

**Joseph Suarez**
jsuarez@mit.edu
Massachusetts Institute of Technology
Cambridge, MA, USA

## Abstract

Most reinforcement learning methods rely heavily on dense, well-normalized environment rewards. DreamerV3 recently introduced a model-based method with a number of tricks that mitigate these limitations, achieving state-of-the-art on a wide range of benchmarks with a single set of hyperparameters. This result sparked discussion about the generality of the tricks, since they appear to be applicable to other reinforcement learning algorithms. Our work applies DreamerV3's tricks to PPO and is the first such empirical study outside of the original work. Surprisingly, we find that the tricks presented do not transfer as general improvements to PPO. We use a high quality PPO reference implementation and present extensive ablation studies totaling over 10,000 A100 hours on the Arcade Learning Environment and the DeepMind Control Suite. Though our experiments demonstrate that these tricks do not generally outperform PPO, we identify cases where they succeed and offer insight into the relationship between the implementation tricks. In particular, PPO with these tricks performs comparably to PPO on Atari games with reward clipping and significantly outperforms PPO without reward clipping.

## 1 Introduction

Reinforcement learning (RL) has shown great promise in addressing complex tasks across various domains. However, each new application typically requires environment-specific tuning and significant engineering effort. The Dreamer [1,2,3] line of work has focused on world-modeling to achieve high performance across a wide range of tasks. The most recent version, DreamerV3, outperforms the previous version with a single set of hyperparameters shared across all tasks. DreamerV3 introduces several stability and performance-enhancing technique to accomplish this, but it also includes several orthogonal changes such as a larger architecture. Additionally, most of these new techniques are not inherently tied to the world-modeling formulation and could potentially be applied in other reinforcement learning settings. Table 1 categorizes the tricks according to which part of the DreamerV3 algorithm they apply to. Specifically, the following techniques are directly applicable to model-free actor-critic methods:

37th Conference on Neural Information Processing Systems (NeurIPS 2023).

- **Symlog Predictions:** A transformation applied to the targets of a neural network, helping to smooth predictions of different magnitudes
- **Twohot Encoding:** A discrete regression objective representing continuous values as a weighting of two adjacent buckets
- **Critic EMA Regularizer:** Regularizes the critic outputs towards its own weight Exponential Moving Average (EMA) to improve stability during training
- **Percentile Scaling:** Scales returns by an exponentially decaying average of the range between their 5th and 95th batch percentile to improve robustness to outliers and varying reward scales
- **Unimix Categoricals:** Combines 99% actor network outputs with 1% uniform random sampling to inject entropy into action selection

Proximal Policy Optimization (PPO)[4] is a popular RL algorithm due to its simplicity and widespread success in a variety of domains. PPO is an actor-critic algorithm that approximates a trust-region optimization strategy to update the policy. The objective function for PPO is given by:

$$L(\theta) = \mathbb{E}_t \left[ \min \left( r_t(\theta) \hat{A}_t, \text{clip}(r_t(\theta), 1 - \epsilon, 1 + \epsilon) \hat{A}_t \right) \right],  \tag{1}$$

where $r_t(\theta) = \frac{\pi_\theta(a_t|s_t)}{\pi_{\theta_{\text{old}}}(a_t|s_t)}$, $\hat{A}_t$ is the estimated advantage function, and $\epsilon$ is a hyperparameter controlling the size of the trust region.

Our work applies the stability techniques introduced in DreamerV3 to PPO, aiming to improve the algorithm's stability and performance. We adapt these techniques for the slightly different algorithmic setting of model-free learning and provide a working implementation tested on various environments. We provide comprehensive ablations demonstrating that these techniques generally do not enhance the performance of PPO. We believe this negative result merits sharing because of the widespread interest in DreamerV3 and serves to temper expectations of the generality of the tricks presented.

The contributions of our work include:

- Applying and adapting DreamerV3's stability techniques to PPO
- Demonstrating the effects of these techniques on PPO's stability and performance across diverse environments
- Analysis of the strengths and weaknesses of each trick, with a more detailed exploration of the two most promising tricks, twohot encoding and symlog predictions
- A high-quality implementation in CleanRL to enable further research in this direction

## 2 Related Work

The Dreamer line of work focuses on learning from imagined experiences using world models. Each version improves upon the previous, with DreamerV3 achieving state-of-the-art results on a variety of tasks. While these algorithms primarily focus on world-modeling, we are interested in model-free algorithms because of their simplicity and widespread applicability. This work focuses on PPO, which has become the primary algorithm for applying RL to new domains in recent years. Numerous extensions and improvements to PPO have been proposed, such as incorporating multiple optimization phases[5] or exploration bonuses[6].

**Table 1:** Application of DreamerV3 Tricks

| Implementation Trick | Actor | Critic | World Model |
|---|---|---|---|
| Symlog Predictions | X | X | X |
| Twohot Encoding | | X | X |
| Percentile Scaling | X | | |
| Critic EMA Regularizer | | X | |
| Unimix Categoricals | X | | X |

Several works have focused on improving the stability of reinforcement learning algorithms. For instance, Pop-Art[7] introduced adaptive normalization of value targets, while R2D2[8] and Ape-X[9] address the challenges of off-policy learning. However, these approaches often require extensive modifications to the base algorithms or focus on specific challenges, whereas our work studies a

set of general stability techniques that are designed to apply more broadly to various reinforcement learning settings.

Previous work has studied the impact of the implementation details of PPO on its performance. Huang et al.[10] described the implementation details of PPO and explored how each one impacts its performance. Engstrom et al.[11] explores how much of PPO's performance over Trust Region Policy Optimization[12] can be attributed to algorithmic improvements versus implementation tricks. None of these discuss the methods explored in our work, though some of the tricks they discuss serve a similar purpose.

In this work, we build on the stability techniques introduced in DreamerV3, exploring their applicability beyond world-modeling. Our approach enhances the performance and stability of PPO when rewards are unbounded. However, as we further show, reward clipping is a simple and strong baseline that is hard to beat. Our work highlights the strengths and weaknesses of these techniques.

## 3 Methods

In this section, we describe the application of the stability techniques introduced in DreamerV3 to the PPO algorithm. We detail each technique and discuss the necessary adaptations for incorporating them into PPO.

We implement these tricks as minimal extensions to CleanRL's extensively validated and benchmarked[13,14] PPO implementation. We used both DreamerV3's open source code base and the paper as references. We found the implementations of all tricks to be consistent with their descriptions in the paper, and we checked several small ambiguities with the authors. We performed manual hyperparameter tuning for each of the implementation tricks, as well as automatic hyperparameter tuning of the learning rate, entropy coefficient, and value loss coefficient using Optuna[15], for the full algorithm with all tricks enabled. We found no improvement in performance and therefore chose to keep CleanRL's default hyperparameters. We also compared our code with all tricks disabled to the original unaltered scripts to ensure that there were no performance regressions as a result of improperly implemented controls on the tricks. Finally, we have open-sourced the implementation of our tricks and experiments https://github.com/RyanNavillus/PPO-v3.

### 3.1 Symlog Predictions

In DreamerV3, symlog predictions are used to compress the magnitudes of target values. Symlog approximates the identity function near the origin so that it has little impact on returns that are already small. The symlog transform and inverse symexp transform are defined in Hafner et al.[3] as:

$$\text{symlog}(x) \doteq \text{sign}(x)\ln\big(|x|+1\big) \qquad \text{symexp}(x) \doteq \text{sign}(x)\big(\exp(|x|)-1\big) \tag{2}$$

We also apply the same transformation to the observations in environments with continuous vector observations and call this method symlog observations.

To use symlog predictions in PPO, we calculate the value loss using the symlog transformation, and we apply the symexp operation to outputs from the critic network exactly the same as in Hafner et al.[3]. When twohot encoding is disabled, for a critic $f(x,\theta)$ with inputs $x$ and parameters $\theta$ we use MSE loss to predict the symlog transformed returns $y$.

$$\mathcal{L}(\theta) \doteq \tfrac{1}{2}\big(f(x,\theta)-\text{symlog}(y)\big)^2 \tag{3}$$

### 3.2 Twohot Encoding

Twohot Encoding is a generalization of one-hot encoding that represents continuous values as a weighting between two equal buckets. For integer values, this is identical to the one-hot encoding of the value. This discrete regression technique allows the critic to predict a distribution over buckets instead of a single value. The definition of twohot encoding from Hafner et al.[3] is shown in Equation 4:

$$\text{twohot}(x)_i \doteq \begin{cases} |b_{k+1} - x| \,/\, |b_{k+1} - b_k| & \text{if } i = k \\ |b_k \quad - x| \,/\, |b_{k+1} - b_k| & \text{if } i = k+1 \\ 0 & \text{else} \end{cases} \qquad k \doteq \sum_{j=1}^{B} \delta(b_j < x) \tag{4}$$

We implement two-hot encoding by replacing the critic value output with 255 bins as in DreamerV3 and calculating the critic value output as the expected value of the softmax distribution of those logits. The critic learns via categorical cross entropy between the critic logits and the twohot encoding of bootstrapped returns from the environment.

As in Hafner et al.[3], when using two-hot encoding we initialize the critic logits to zero to prevent it from predicting extremely large values at the start of training. In ablations with symlog enabled, we follow DreamerV3 and set the range of the twohot bins to [-20, 20], allowing the critic to represent values from $[-\exp(20), \exp(20)]$. When symlog is disabled, we instead choose a range of $[-15000, 15000]$ for Atari environments without reward clipping enabled, and $[-1000, 1000]$ for Atari environments with reward clipping enabled as well as the DeepMind Control Suite. These ranges were chosen to allow the critic to represent the maximum and minimum values seen during training. We find that the choice of range for two-hot encodings has a significant impact on performance. We include limited experiments demonstrating this relationship in subsection 4.5

### 3.3 Critic EMA Regularizer

Hafner et al.[3] regularize the critic towards an Exponential Moving Average (EMA) of its own outputs, improving stability during training.

We use the critic EMA to regularize the critic loss using the same decay rate (0.98) and regularizer coefficient (1.0) as DreamerV3. We tried tuning these values but found little difference in performance. The EMA is updated once during each optimization step. For our ablation studies, when two-hot encoding is enabled, we use categorical cross-entropy loss between the critic logits and EMA logits. When two-hot is disabled, we calculate the Mean Squared Error (MSE) between the critic and EMA value predictions.

### 3.4 Percentile Scaling

As discussed in Hafner et al.[3], previous work on PPO has typically scaled returns by dividing them by their exponentially decaying standard deviation. However, they point out that when rewards are sparse and the standard deviation is small, this method amplifies the noise in small returns, preventing the policy from exploring sufficiently. Percentile scaling instead tracks an exponentially decaying average of the 5th and 95th percentile of each batch of returns. Scaling advantages by the difference between these percentiles promotes exploration while allowing DreamerV3 to use a fixed entropy scale. Instead of directly scaling returns as in DreamerV3, we scale the advantages predicted by Generalized Advantage Estimation. As in Hafner et al.[3] we only scale advantages when the scaling factor (the difference between the 5th and 95th percentile) is greater than 1 to avoid amplifying errors in small advantages.

To add this method to PPO, we compute bootstrapped returns in the same way as DreamerV3, then scale the advantages used to calculate the policy loss. Note that we do not modify the values or returns used to calculate the critic loss. We also found that when the EMA updated too quickly, it could lead returns to smoothly drop to 0 in the DeepMind Control Suite. To counteract this, we change the percentile EMA decay rate from 0.99 to 0.995 for the DeepMind Control Suite and 0.999 for the Arcade Learning Environment. We also found that in practice this method works best in combination with the standard advantage normalization used in PPO.

### 3.5 Unimix Categoricals

Unimix categoricals are a mixture of 99% neural network outputs and 1% random uniform sampling used for action selection. This prevents the probability of selecting any action from being zero, and encourages exploration similar to an entropy bonus. Unimix categoricals are only used in the Atari experiments because the DeepMind Control Suite does not use categorical actions. We do not experiment with changing the unimix ratio in this paper and use 1% as in Hafner et al.[3].

## 4 Experiments

In the following sections, we explain the environments we use to evaluate the implementation tricks. We test each trick on the entire environment suite across multiple seeds. We used approximately

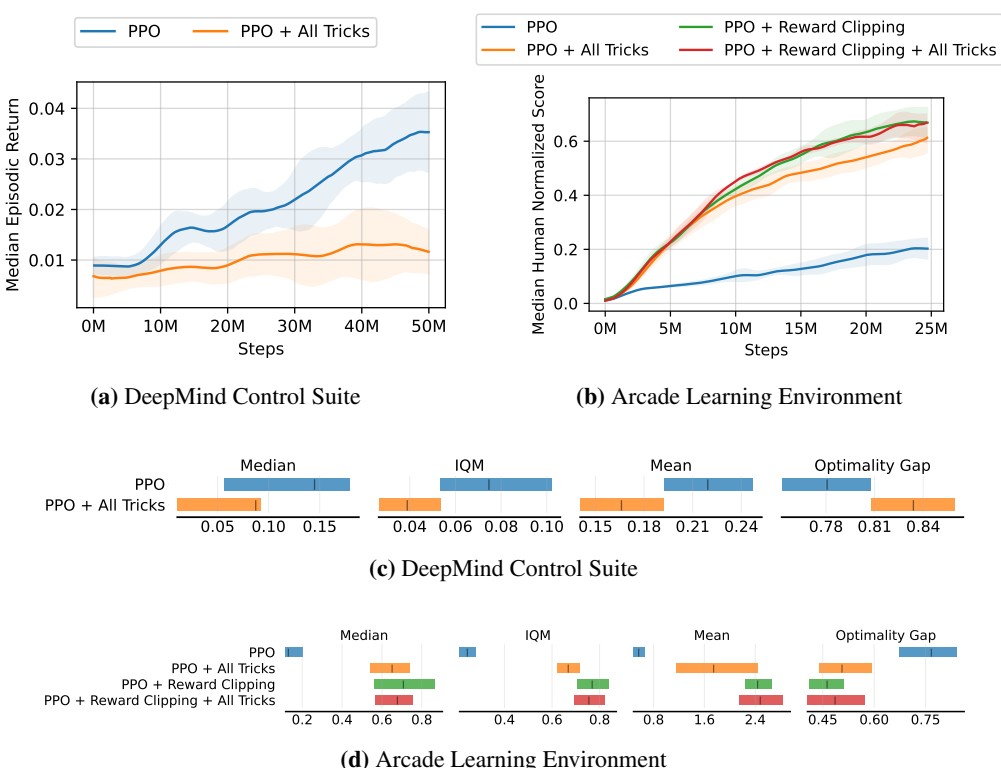

**(a)** DeepMind Control Suite

**(b)** Arcade Learning Environment

**(c)** DeepMind Control Suite

**(d)** Arcade Learning Environment

**Figure 1:** PPO with all tricks enabled compared to base PPO, using the DeepMind Control Suite in a) and c) and Atari environments (with and without reward clipping) in b) and d).

8000 GPUs hours for the experiments in this paper as well as 4000 more for testing and development, most of which were run on Nvidia A100s. We report results using standard metrics as well as those provided by the RLiable library as recommended in Agarwal et al.[16] when results require uncertainty measures. They describe a methodology and metrics for creating reproducible results with only a handful of runs by reporting uncertainty. These metrics include 95% stratified bootstrap confidence intervals for the mean, median, interquartile mean (IQM), and optimality gap (the amount by which an algorithm fails to meet a minimum normalized score of 1).

## 4.1 Environments

### 4.1.1 Atari 100M

We train agents on each of the 57 environments in the Arcade Learning Environment[17] for multiple seeds in each of our ablations. In this paper, we are examining whether these tricks improve PPO's robustness to different reward scales. The high scores, or maximum possible returns, for Atari games range from 10 for Surround to 89 million for VideoPinball. They also have drastically different reward densities, where some games guarantee a reward at each step while others require the agent to learn complex behavior to experience any positive reward. Typically, previous work on the Arcade Learning Environment has used reward clipping to limit individual step returns to 1[18]. This significantly reduces the scale of returns and increases the effective density of rewards by weighting all nonzero rewards equally. To better study different reward scales, we perform ablations with and without reward clipping enabled. Aside from reward clipping, we use the standard wrappers recommended by Machado et al.[19]. For Atari experiments, we standard benchmark of median human normalized scores[20] instead of episodic returns.

### 4.1.2 DeepMind Control Suite

We also test our method on 35 proprioceptive control environments from the DeepMind Control Suite. These are physics-based environments with well-normalized reward functions. Each environment has

a minimum return of 0 and a maximum return of 1000, allowing us to test how these tricks perform on environments with returns that are already well-normalized. Rewards can also take on decimal values unlike in Atari environments. In our plots, we divide returns by 1000 to limit them to the range [0, 1] and ensure that optimality gap has a consistent definition across environments.

## 4.2 Enabling All Tricks

We first compare our version of PPO with all of the stability tricks enabled to the PPO baseline in each set of environments in Figure 1.

When reward clipping is enabled, we see that PPO achieves similar performance with and without the tricks. When reward clipping is disabled, the stability tricks allow PPO to recover most of the performance of PPO with reward clipping. This suggests that the tricks make PPO significantly more robust to varying reward scales, though they slightly underperform a simple reward clipping baseline.

## 4.3 Add-One Ablations

We perform add-one ablations where we enable one trick at a time and evaluate on both the DeepMind Control Suite and Atari environments (with and without reward clipping) to determine if any tricks provide a general improvement to the PPO baseline. The results are shown in Figure 2.

We find that all of the tricks perform comparably or slightly worse than the PPO baseline on the DeepMind Control Suite and Atari with reward clipping. In particular, symlog predictions and twohot encoding underperform compared to the baseline. However, we see that symlog predictions dramatically improve performance when reward clipping is disabled, and all of the remaining tricks perform comparably or better on most metrics. This indicates that the tricks are effective as return normalization tools, but underperform when returns are already normalized to reasonable ranges.

## 4.4 Drop-One Ablations

We perform drop-one ablations where we enable all tricks and disable one at a time to see the interactions between tricks and identify combinations that might outperform PPO. The results are shown in Figure 3. In these Atari experiments, reward clipping is disabled when symlog is enabled. Our drop-one ablations focus on symlog predictions, twohot encoding, percentile scaling, and the critic EMA regularizer. We exclude unimix categoricals from the Atari drop-one ablations because they had little impact in the add-one experiments and do not interact with the other tricks. Likewise, we exclude symlog observations from the DeepMind Control Suite drop-one ablations because they do not interact with the other tricks.

We find that for the DeepMind Control Suite, removing twohot encoding and the critic EMA regularizer actually improves performance. Removing percentile scaling and symlog predictions seems to harm performance while in the Arcade Learning Environment, only removing symlog predictions harms performance. In our add-one ablations for both environment suites, adding symlog predictions and twohot encodings to PPO also performed worse than the baseline. Our twohot encoding experiments require us to define a single bin value range across each entire environment suite which may be causing its lackluster performance. To diagnose this issue, we explore the interaction between symlog predictions and twohot encoding in the following section.

## 4.5 Symlog Predictions and Twohot Encoding

Twohot encoding can only represent a bounded range represented by its bins. In Atari environments with widely-ranging possible scores, it can be difficult or impossible to choose tight bounds, which is why it is always paired with a symlog transform in Hafner et al.[3]. This section provides additional context to the interactions between these tricks and the results are displayed in Figure 4. We first examine the effects that the range and number of bins have on the performance of PPO with twohot encoding. Large bounds also seem to have a detrimental effect on learning, possibly by reducing the effective number of bins used to predict values. We see that mean episodic return increases with the number of bins, then falls off at a much slower rate as we increase past the optimal number. Surprisingly, the agent only suffers a small drop in performance even when the twohot range is set to 1. It's possible that the performance loss would be more significant in an environment with larger

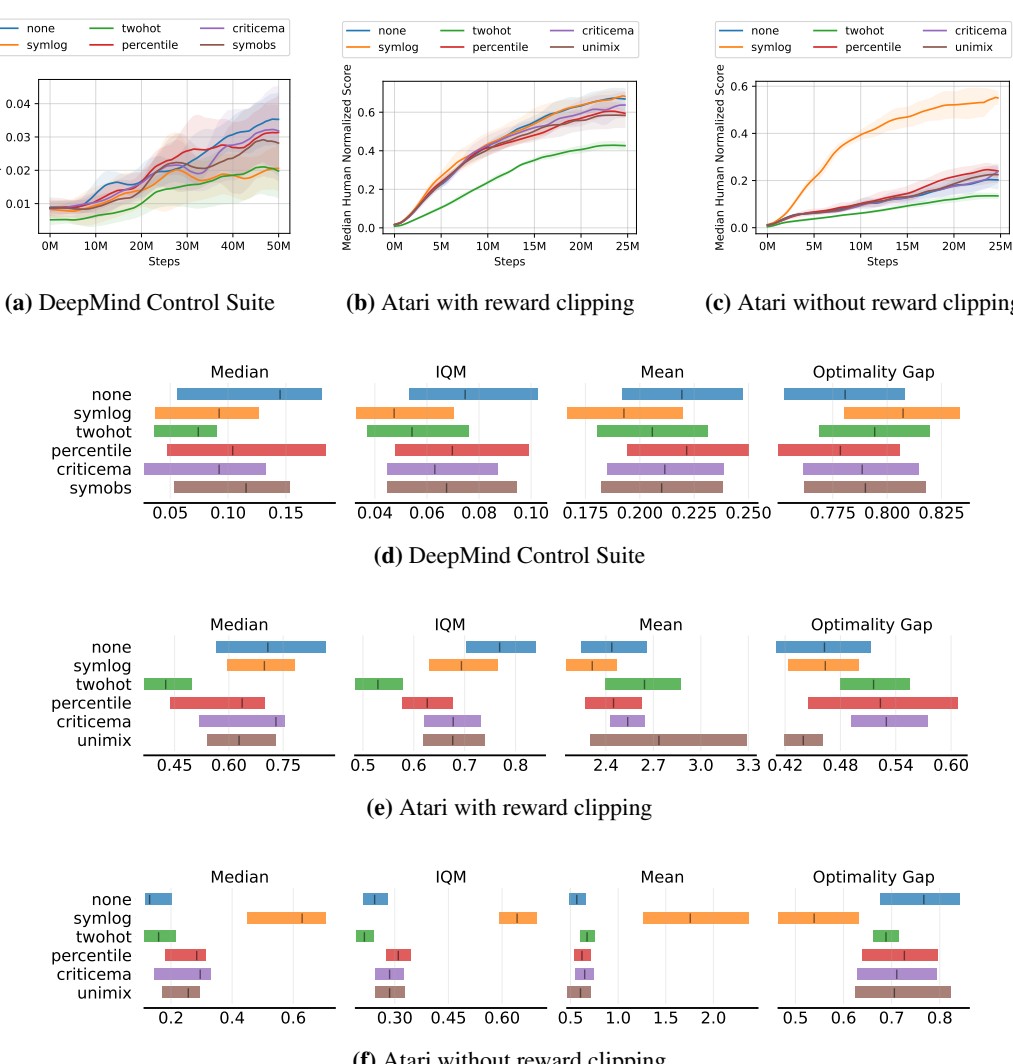

**(a)** DeepMind Control Suite   **(b)** Atari with reward clipping   **(c)** Atari without reward clipping

**(d)** DeepMind Control Suite

**(e)** Atari with reward clipping

**(f)** Atari without reward clipping

**Figure 2:** Median scores averaged across multiple seeds for the add-one ablations in each environment set. 5 seeds were used for Deepmind Control Suite and 3 seeds for Atari environments.

returns than Breakout where the critic value prediction is much farther from the true value. We also see in Figure 4c that even when combined with symlog predictions, twhot encoding underperforms compared to base PPO across the entire Arcade Learning Environment.

## 5   Discussion

The tricks tested in this paper allow PPO to achieve strong performance in environments with drastically different reward scales using a single set of hyperparameters, while the original PPO algorithm is unable to do so. Our modified version of PPO performs comparably to the original algorithm in Atari games with and without requiring reward clipping, but we see that our agents perform worse on the DeepMind Control Suite. This suggests that the tricks may be poorly suited to environments with normalized, bounded, continuous rewards. Symlog predictions are by far the most impactful trick in all experiments. Conversely, all of our experiments seem to suggest that twohot encoding is a detrimental addition to PPO even in combination with other tricks. Percentile scaling, the critic EMA regularizer, and unimix categoricals all slightly improve performance when we disable reward clipping, and slightly harm performance when we enable it, again suggesting that they are most useful in environments without normalized returns. Symlog observations underperform in the DeepMind Control Suite, but could be useful in environments with larger unbounded observations.

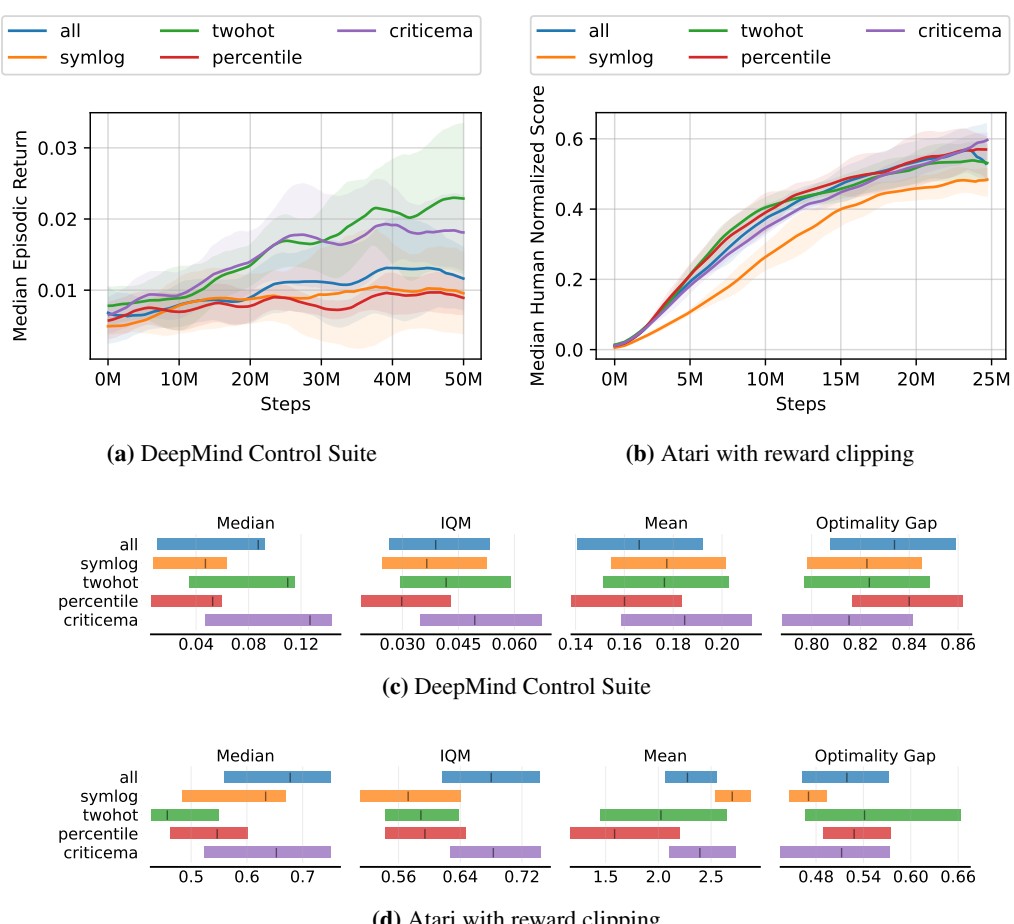

**(a)** DeepMind Control Suite

**(b)** Atari with reward clipping

**(c)** DeepMind Control Suite

**(d)** Atari with reward clipping

**Figure 3:** Median scores for the drop-one ablations in each environment set. 5 seeds were used for DeepMind Control Suite and 3 seeds for the Arcade Learning Environment

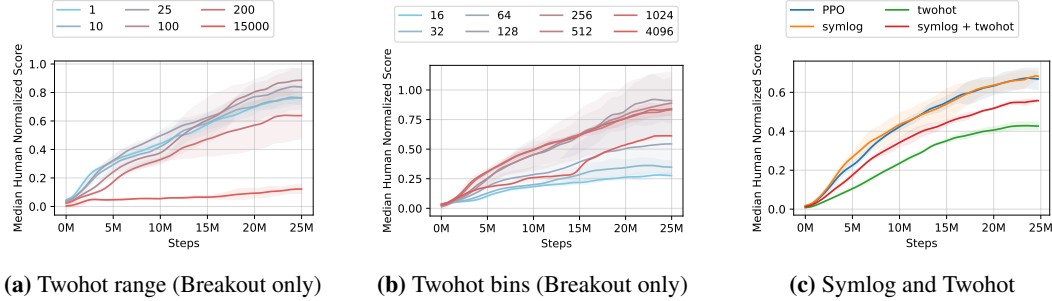

**(a)** Twohot range (Breakout only)     **(b)** Twohot bins (Breakout only)     **(c)** Symlog and Twohot

**Figure 4:** Experiments varying the a) range (bins=256) and b) number of bins (range=100) used for twohot encoding in Atari Breakout, averaged across 3 seeds. c) shows symlog predictions with twohot encoding across all 57 games in the Arcade Learning Environment.

In this work, we applied stability tricks introduced by DreamerV3 to PPO and demonstrated that they do not result in a general performance improvement, but can be valuable in specific cases. The fact that these tricks do not benefit PPO raises the question of how exactly they impact DreamerV3. These tricks are directly applicable to PPO and required little to no modification to implement. It is possible that they may work for the entropy-regularized actor-critic that DreamerV3 uses, which is likely a weaker baseline than PPO on its own. It's also possible that symlog predictions, twohot encoding, or unimix categoricals are specifically beneficial to world model learning, or that the world model-specific tricks not studied in this paper are the source of its improved performance. We note

that their use of SiLU activation functions[21], layer normalization[22], and a 200 million parameter architecture may also contribute to its state-of-the-art results. We include one experiment using a similar architecture in the appendix, but find that it does not work well for PPO.

Hafner et al.[3] make few direct comparisons to the previous DreamerV2 method and include limited ablations on only six environments, so it is difficult to confirm exactly how each trick contributes to performance. The ablation studies in this paper should serve as a valuable reference for studying these tricks and developing new methods that promote robustness to varying reward scales in reinforcement learning. Solving this problem would allow RL to more easily be applied to new problems without extensive engineering efforts. DreamerV3 achieves state-of-the-art results, but our results show that the relationship between these implementation tricks and those results may not be as straightforward as it would seem otherwise. Due to the importance of this problem and the impressive results of DreamerV3, we believe these tricks deserve further investigation both using DreamerV3 and in new contexts.

## 6  Limitations

Due to the many differences between PPO and DreamerV3, we cannot say whether the findings in this paper transfer to DreamerV3 or other similar algorithms. Our experiments also do not fully cover the range of tasks evaluated in the original DreamerV3 paper. We have focused on a more limited set of environments in order to provide thorough, high-quality ablations. We experiment on Atari environments without reward clipping to study the effects of each trick on poorly normalized reward scales, and the DeepMind Control Suite for environments with well-normalized returns, but it's possible that we would see different results on more complex environments.

## 7  Conclusion

We have presented a thorough study of the stability techniques introduced in DreamerV3 to the widely used PPO algorithm. These experiments allow us to identify key areas where tricks improve performance and demonstrate the broader applicability and potential benefits of these techniques for the reinforcement learning community. In the spirit of openness and reproducibility, we have released our complete code base and experiment data at https://github.com/RyanNavillus/PPO-v3, further promoting the adoption and study of these techniques in the reinforcement learning community.

## 8  Acknowledgments

We would like to thank James MacGlashan for his helpful comments and suggestions, as well as CarperAI for providing the majority of the compute used for our experiments.

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
