# 1 Appendix A: Atari Add-One Ablations

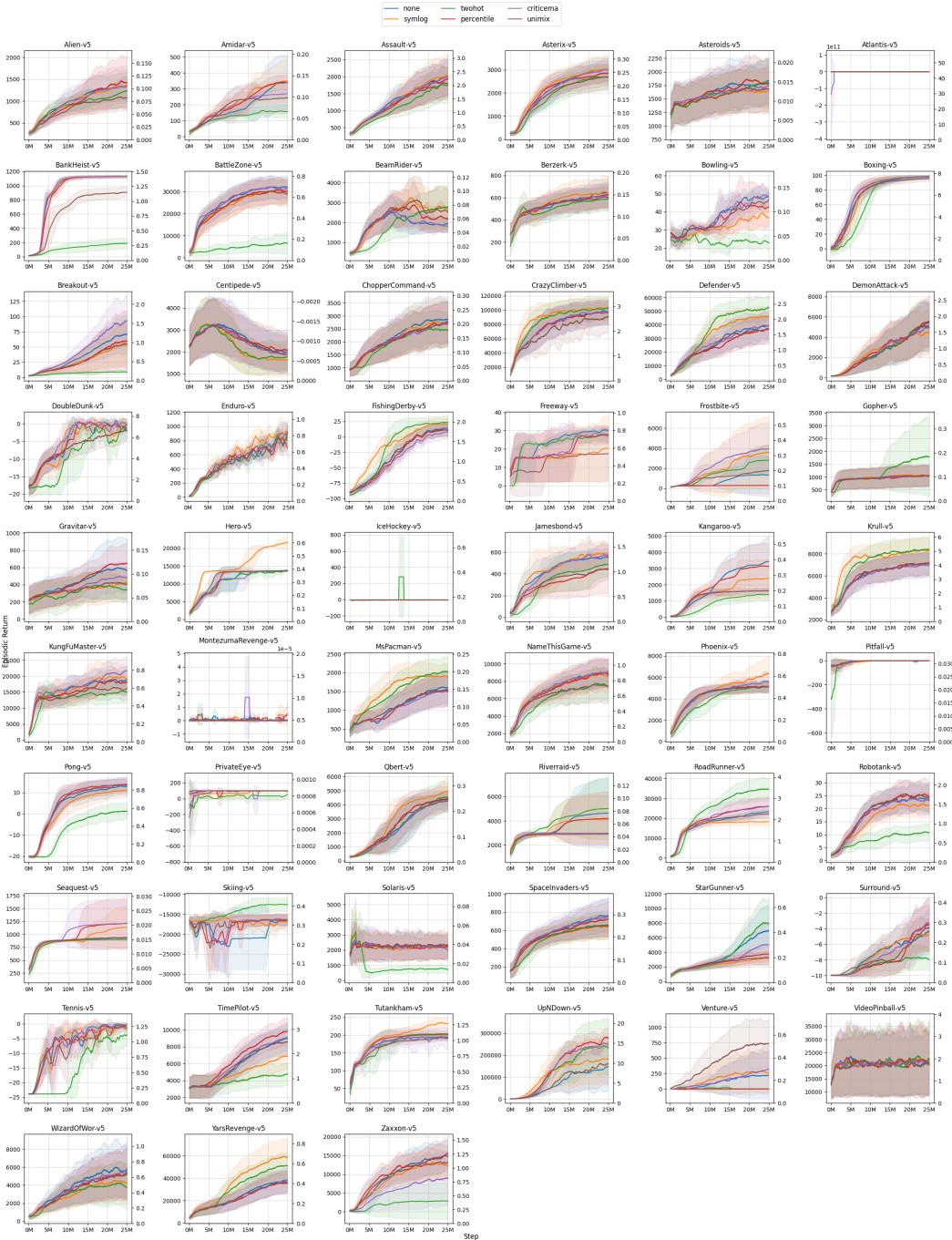

Figure 1: Add-one ablations for the Arcade Learning Environment with reward clipping. Each line shows the median scores for 57 environments, averaged over 3 seeds, across training.

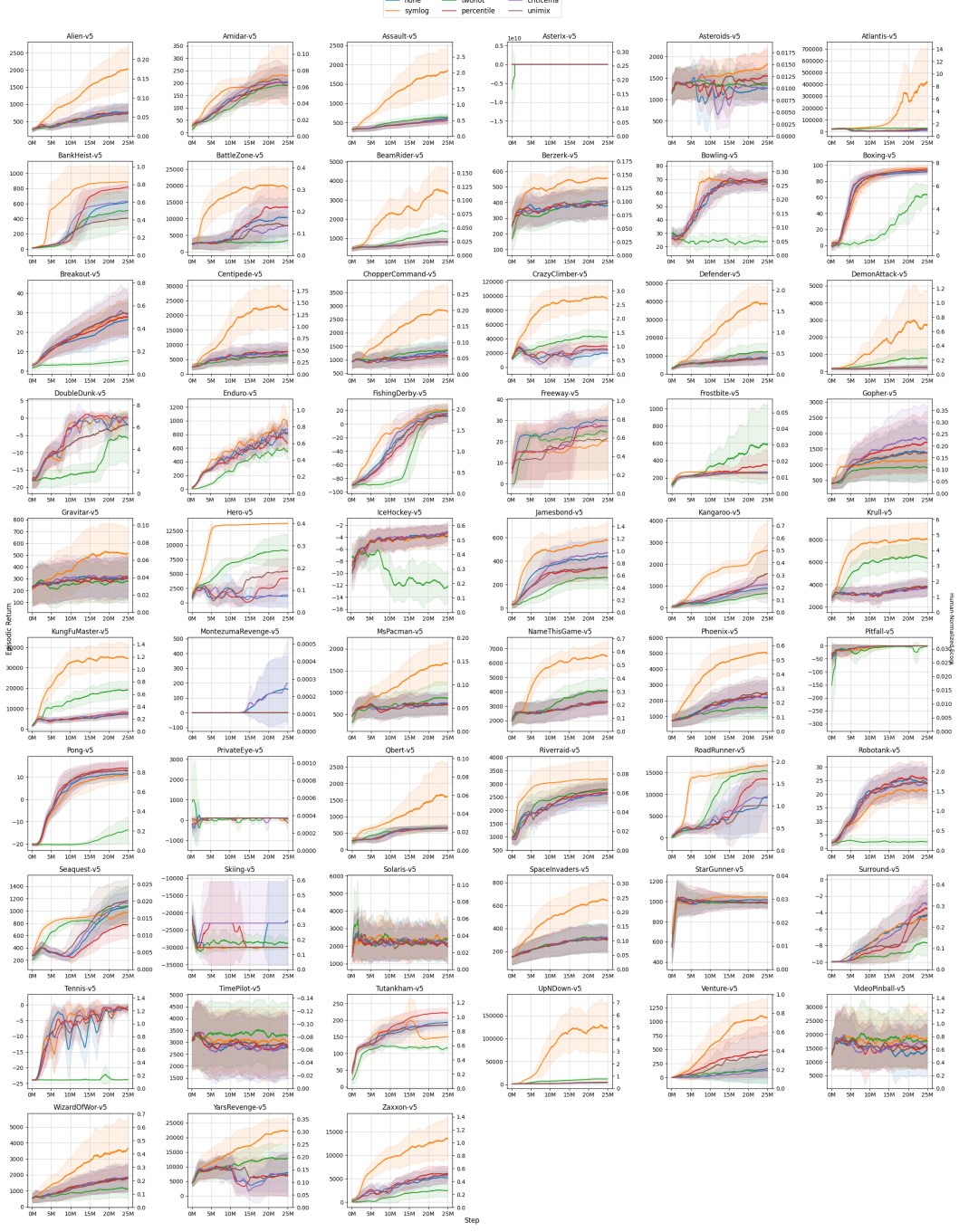

Figure 2: Add-one ablations for the Arcade Learning Environment without reward clipping. Each line shows the median scores for 57 environments, averaged over 3 seeds, across training.

# 2 Appendix B: Atari Drop-One Ablations

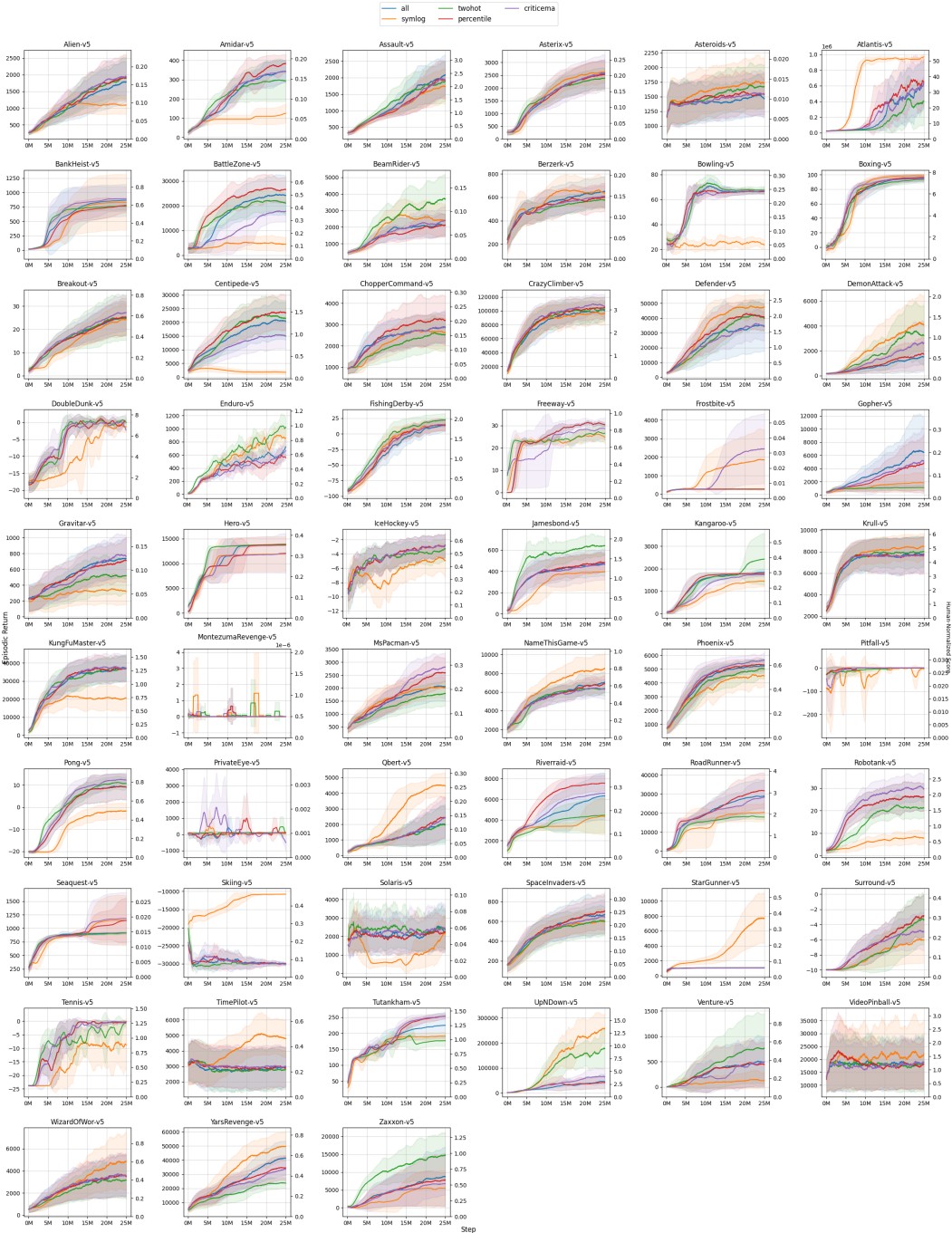

Figure 3: Drop-one ablations for the Arcade Learning Environment with reward clipping. Each line shows the median scores for 57 environments, averaged over 3 seeds, across training.

# 3 Appendix C: DeepMind Control Suite Add-One Ablations

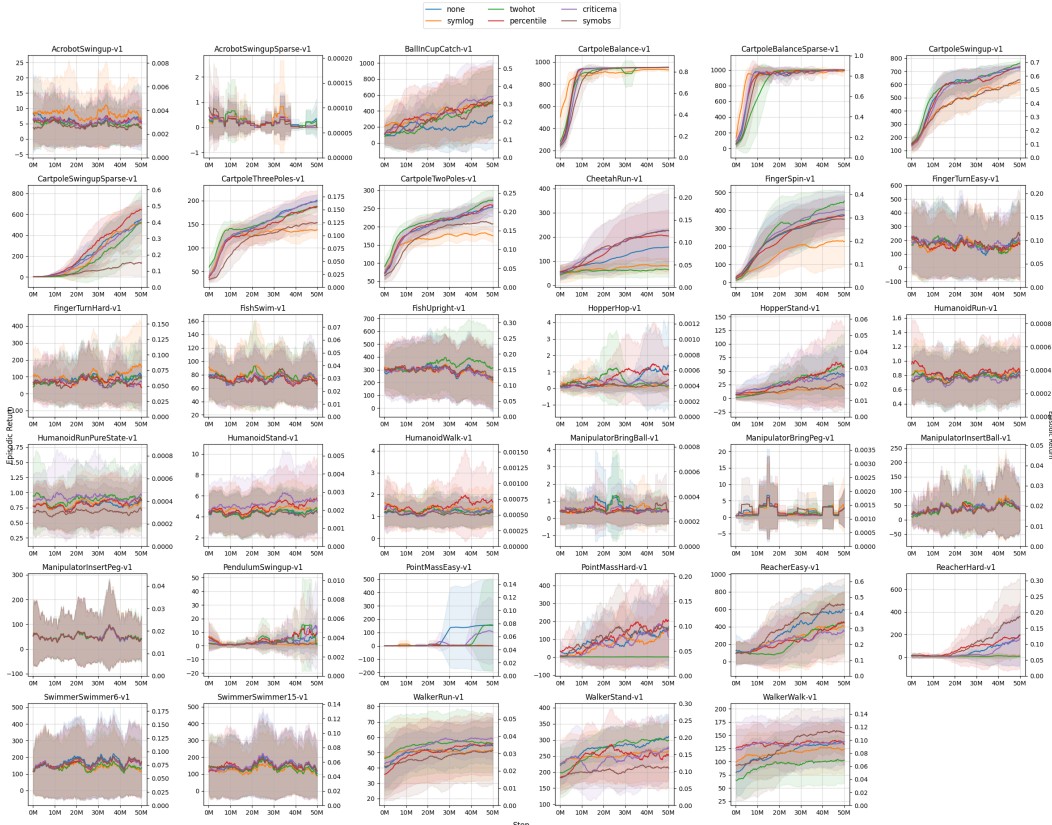

Figure 4: Add-one ablations for the DeepMind Control Suite. Each line shows the median scores for 35 environments, averaged over 5 seeds, across training.

# 4 Appendix D: DeepMind Control Suite Drop-One Ablations

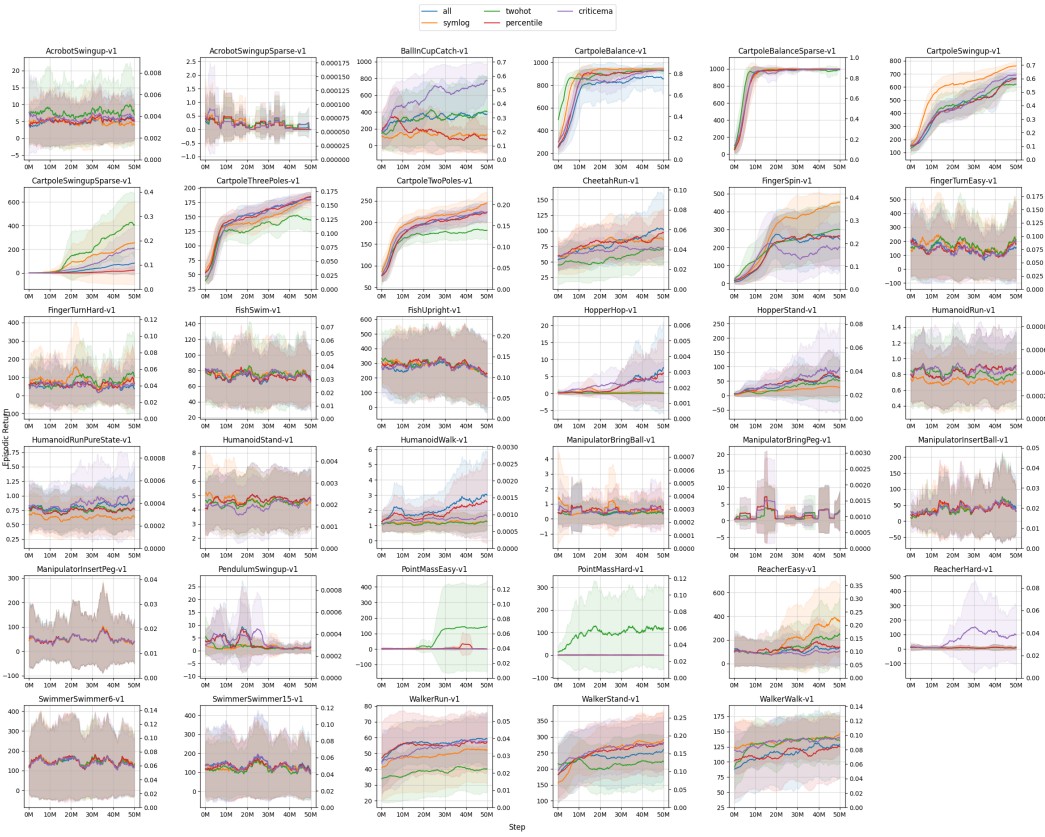

Figure 5: Drop-one ablations for the DeepMind Control Suite. Each line shows the median scores for 35 environments, averaged over 5 seeds, across training.

# 5 Appendix E: Architecture

We use the 20 million parameter XL DreamerV3 encoder and actor-critic architecture in PPO and compare its performance to the 1 million parameter Nature CNN used in the rest of our experiments.

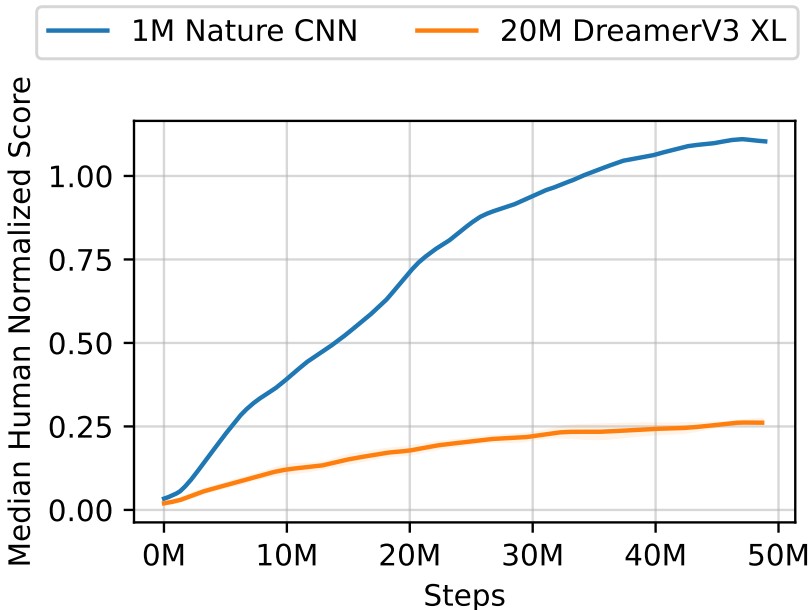

Figure 6: 20M DreamerV3 XL architecture vs. 1M Nature CNN. Figures show median performance averaged across 3 seeds and all 57 Atari environments.

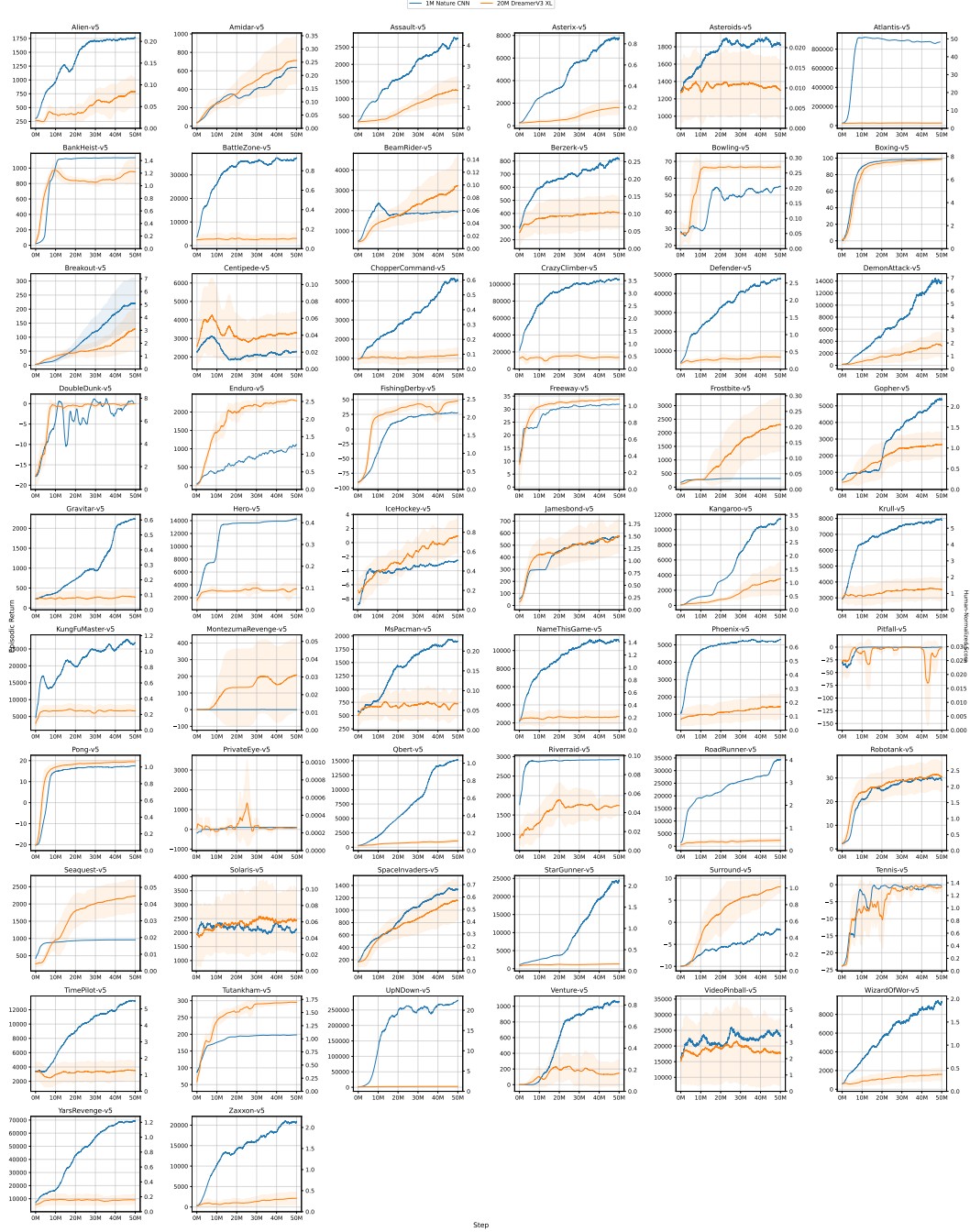

Figure 7: 20M Dreamer-v3 XL architecture (3 seeds) vs. 1M Nature CNN (1 seed).

 # 6   Appendix F: PPO Hyperparameters

Table 1: The default PPO hyperparameters in CleanRL's implementation, used for all experiments. The only change we make is increasing the number of environments for better time efficiency.

| | |
|---|---|
| Learning Rate | $2.5e^{-4}$ |
| Environments | 128 |
| Steps | 128 |
| $\gamma$ | 0.99 |
| GAE $\lambda$ | 0.95 |
| Minibatches | 4 |
| Epochs | 4 |
| Clip coefficient | 0.1 |
| Value loss clipping | Enabled |
| Entropy Coefficient | 0.01 |
| Value loss coefficient | 0.5 |
| Max Grad Norm | 0.5 |