# OpenReview forum: "Reward Scale Robustness for Proximal Policy Optimization via DreamerV3 Tricks"
_NeurIPS.cc/2023/Conference — NeurIPS 2023 poster_

### Official Review · Reviewer_QvZz · 2023-07-03

**Soundness:** 3 good
**Presentation:** 4 excellent
**Contribution:** 3 good
**Rating:** 7
**Confidence:** 5

**Summary:**

This paper explores implementation details from Dreamer-v3 in the context of PPO to examine the generality of these implementation tricks to DeepRL. They identify 5 unique implementation details from Dreamer-v3 and do a thorough examination of its effects on both continuous and discrete control environments (i.e. Deepmind Control Suite and Atari respectively). They offer solid evidence for and against certain implementation tricks in the context of PPO and find that environments with poorly normalized, unbounded rewards were where these identified tricks helped most.

**Strengths:**

### Originality
---
This paper follows a line of literature that investigates implementation tricks for DeepRL algorithms. I believe its valuable and necessary research given the large amount of implementation tricks in DeepRL. It was interesting and original for the authors to specifically study tricks from an on-policy model-based DeepRL algorithm to the model-free DeepRL algorithm.

### Quality and Clarity
---
I believe they did a very thorough evaluation on very large benchmark domains. It is impressive that they were able to do both all of Atari and DMC control. The writing was clear and I believe the authors were very clear about what they were investigating and presenting from the very beginning.

### Significance
---
The per-environment learning curves from the ablations and experiments for both DMC and Atari (in the supplementary) is a significant contribution that democratizes this information to many researchers who do not have great compute access. Although there wasn't a strong clear implementation trick that drastically improved performance, across DMC and Atari, I commend the authors for reporting all both what worked well and not well which is very useful for practitioners moving forward.

**Weaknesses:**

* It would be great to see hyperparameter ranges tested for each implementation detail. Perhaps this is already done through the code-release but would be nice to have in writing with some insight from the authors
* Perhaps diving a bit deeper into subcategories of tasks within the benchmark suites and doing some analysis there. For example, some environments are long horizon, sparse/dense reward, small/large action space, etc. Given all of the per-environment data already documented and available, it would be great if the authors could provide some insight on why some implementation tricks worked/didn't work based upon features of the task.
* Some more insight on which implementation tricks were more tailored towards World Model learning and specifically Dreamer's actor-critic learning. It would be interesting to implement these implementation tricks with A2C rather than PPO to test whether these tricks were indeed for the world model or for the RL optimization. (I would suggest PPO for Dreamer-V3 but that also sounds very non-trivial).

**Questions:**

My questions align with the points raised in the "weaknesses" section:
* Other than the reward clipping ranges, did you find these implementation tricks very sensitive to hyperparameter?
* Where there other insights about types of environments within a benchmark suite and its relationship to the implementation tricks?
* What was the main rationale for choosing PPO over A2C when applying Dreamerv3 tricks? How many of implementation details do you think were specific to World Model training?

**Limitations:**

Yes, they have adequately addressed the limitations.

---

> ### Author Rebuttal · Authors · 2023-08-09
>
> Thank you for taking the time to thoroughly consider our work. Based on your feedback, we will make the following clarifications and additions to the paper:
>
> Hyperparameters: Symlog does not have any hyperparameters and we don’t experiment with changing the unimix ratio from 0.01. The Critic EMA did not seem particularly sensitive to its hyperparameters in our tests.
>
> In our paper, we already explore the hyperparameters for two-hot (number of bins and the range of values represented by those bins) in section 4.5. We also mentioned in section 3.4 that percentile scaling was very sensitive to its EMA decay rate, and we had to increase it slightly from the default in DreamerV3 to avoid some learning regressions. We will explain that this may be related to the fact that DreamerV3 and PPO calculate returns at different points, so they likely update this EMA at different rates.
>
> Task Subcategories: We did not observe any performance correlations with the well-known taxonomies of Atari games (easy/hard exploration, dense/sparse rewards). There also does not appear to be a correlation with reward sparsity in individual DMC environments. It seems that in general symlog predictions perform better on environments with relatively large reward ranges, but not as consistently as one might expect. While there is no clear property that predicts the performance of the implementation tricks, we hope others will find the plots to be valuable references for their own work.
>
> World Models: Free bits and the world model regularizer introduced by DreamerV3 are only applicable to model-based methods. The remaining tricks apply to some components of actor-critic learning. It is also possible that some of the actor-critic tricks perform better in a model-based system. In particular, the percentile scaling and critic EMAs might have a greater effect when returns are generated by the model instead of the environment. We have no reason to suspect that two-hot encoding, symlog predictions/observations, or unimix categoricals would perform better in a model-based context. Unfortunately testing this explicitly would require extensive changes to the DreamerV3 codebase or more likely a full reimplementation.
>
> Question: We do not fully understand your suggestion of using A2C, given that it is just a special case of PPO with no sample reuse and some minor implementation detail differences. Is your intent that we should run on a simpler RL algorithm without the bells and whistles of PPO? Or did you perhaps mean to suggest a different algorithm that includes world modeling?

---

> > ### Comment · Reviewer_QvZz · 2023-08-10
> >
> > Hi, I have read the rebuttal. Thank you for taking the time to answer my questions! To clarify my question about A2C, I was wondering about the effect of algorithm choice on the policy gradient front given Dreamer-v3 used a policy gradient closer to the unclipped PPO (like A2C) rather than the PPO objective used. More of an intermediate ablation to investigate the design choices tested in this work.

---

> > > ### Author Response · Authors · 2023-08-15
> > >
> > > It's possible we might see a greater effect on A2C or on REINFORCE, which is the policy gradient method used by Dreamer. That being said, if the tricks help either of those methods but not PPO, it might suggest that they provide redundant improvements to PPO's own implementation tricks. It would have been an interesting question to answer if we had more time or compute, so thank you for the suggestion!

---

### Official Review · Reviewer_WmdD · 2023-07-04

**Soundness:** 4 excellent
**Presentation:** 4 excellent
**Contribution:** 2 fair
**Rating:** 7
**Confidence:** 5

**Summary:**

This paper aims to assess the generality of DreamerV3's stabilization techniques by adapting them to the widely-used model-free algorithm PPO. The authors then proceed to conduct a thorough ablation study across Atari games and the DeepMind Control Suite to evaluate their effectiveness.

**Strengths:**

- The paper is well written and easy to follow.
- The obtained findings are intriguing and highlight the need for a more comprehensive ablation study of these tricks in the model-based context of DreamerV3.
- Demonstrating a negative result poses a challenge as poor outcomes can be attributed to implementation errors. In this regard, the authors have done a rather good job:
    - They chose PPO, an extensively studied algorithm, and conducted experiments based on the widely-accepted CleanRL implementation.
    - They perform add-one and drop-one ablations across the complete set of DMC environments and Atari games, while adhering to the recommended Atari wrappers. Multiple seeds were used, and the results were reported using the recommended metrics from RLiable.
    - They rely on the open source implementation of DreamerV3 and communicated with DreamerV3’s authors to implement theses tricks. The few differences with DreamerV3 are well-explained and justified.
    - The code is open source, which enables the community to further validate their study and potentially build upon it.

**Weaknesses:**

- Originality: The sole innovation of the paper lies in the adaptation of these tricks to PPO, which, on its own, may not be considered a significant contribution.
- Impact: While the experiments demonstrate technical rigor and the publication of negative results is valuable, the overall impact of the paper appears to be relatively limited.
- Minor writing details:
    - There are instances in the bibliography where names with accents have been incorrectly rendered.
    - Some references should probably be included (L120, relevant papers on Twohot encoding for distributional RL; L148, explicit the previous work, etc).

**Questions:**

The paper is coherent and does not present any confusion, so I don't have any specific questions. I am open to discussing the impact further, as it is undoubtedly challenging to evaluate in advance.

**Limitations:**

The authors discussed the scientific limitations of their work. The societal impact is not discussed, but not very relevant in this case I believe.

---

> ### Author Rebuttal · Authors · 2023-08-09
>
> Thank you for your detailed review of our work. We have no objections and will fix the minor writing details, thank you for pointing those out. As a point of discussion, we would like to add some context as to the impact of our work. The DreamerV3 paper was received with significant and sustained academic interest. Every other RL researcher we spoke to believed that the algorithmic tricks introduced in the paper would be immediately applicable to other algorithms. Our work indicates that this is not the case. As you mentioned, negative results require extreme care in implementation, and this study required a combination of extensive testing, code review by all of the authors, and correspondence with the DreamerV3 authors. Finally, we’d like to emphasize that a major contribution of this work is the extensive ablation studies, as well as a detailed study of two-hot encoding and symlog predictions specifically, not only their adaption to PPO.
>
> If the above changes your impression of the importance of this analysis and the difficulty and expense of conducting it, please consider updating your score.

---

> > ### Comment · Reviewer_WmdD · 2023-08-14
> > **Thanks for the rebuttal - Updating to 7**
> >
> > I thank the authors for their response.
> >
> > I concur with reviewer 75kB's suggestion that including a comparison with DreamerV3's results would serve as an effective sanity check. However, I understand that this could be challenging due to the different codebases used for PPO and DreamerV3.
> >
> > Since implementation details are crucial in deep RL, I believe investigating transferability and reproducibility is important and should be encouraged. I'm changing my score from 6 to 7.

---

> > > ### Author Response · Authors · 2023-08-15
> > >
> > > Thank you for your comments and for adjusting your score! We agree that certain experiments with the DreamerV3 codebase could have been valuable if we were able to conduct them. We hope that the disparity between the effectiveness of these tricks for PPO and DreamerV3 will encourage further investigation from the community.

---

### Official Review · Reviewer_DsH5 · 2023-07-04

**Soundness:** 3 good
**Presentation:** 2 fair
**Contribution:** 2 fair
**Rating:** 3
**Confidence:** 4

**Summary:**

This paper empirically tests the effectiveness of the DreamerV3 tricks on PPO. The authors conduct a comprehensive study on each trick, especially, extensive ablation studies, including add-one and drop-one ablations. Based on these experiments, the authors give some insights into the effect of the DreamerV3 tricks.

**Strengths:**

- Extensive empirical study on each of the DreamerV3 tricks;
- Experiments are conducted on various environments, including the environments from Atari 100M and DeepMind Control Suite, and using the rliable metrics to present robust performance comparisons;
- Based on the experiments, the authors give some insights.

**Weaknesses:**

- The largest weakness of this paper is that they test the effectiveness of DreamerV3 tricks on only a single RL algorithm, i.e., PPO. Given the diversity of RL algorithms, the conclusions derived on PPO, at a large possibility, cannot generalize to other RL algorithms. Indeed, the observations given by this paper are valuable for the community as a technical report, but I don't think the contributions of this paper are enough for a NIPS paper.
- The authors claim that "our method performs comparably to PPO on Atari games with reward clipping and significantly outperforms PPO without reward clipping". However, it can be seen from Figure 2 (e) and (f) that, the algorithms with reward clipping generally outperform the ones without reward clipping, which suggests that reward clipping is a useful technique. Therefore, I don't think it is valuable to achieve a SOTA performance under the condition without reward clipping. The vanilla PPO with reward clipping is still the SOTA method on Atari.
- Furthermore, on DeepMind Control Suite, the vanilla PPO is also the SOTA algorithm. So it can be concluded that almost all the DreamerV3 tricks do not have a positive effect on PPO, leading the conclusions of this paper to be much less meaningful.

**Questions:**

- Do the observations of this paper applicable to other RL algorithms?

**Limitations:**

The authors have adequately addressed the limitations and potential negative societal impact of their work.

---

> ### Author Rebuttal · Authors · 2023-08-09
>
> Thank you for your time and effort in reviewing our work. We certainly agree that there are areas where our focus on PPO limits generalization to other RL algorithms, but we believe it is important to emphasize the significance of PPO as an algorithm. There is a large body of work aimed at studying and improving PPO specifically because it is the most popular RL algorithm [1] [2]. Our work transferring techniques from a very well known model-based RL algorithm to the most popular model-free algorithm will be of significant interest to many RL practitioners, especially since every RL researcher we talked to was sure that they would work. We explain our choice to use PPO further in our overall response.
>
> The purpose of experimenting on Atari without reward clipping was to study a case with widely varying reward scales, which is where these tricks were expected to help most. We see in this case that symlog significantly improves PPO’s base performance here, as well as percentile scaling to a lesser degree. We see in our experiments with Atari and DMC that symlog is actually valuable in environments with large reward ranges, but less effective when rewards are already well normalized. These tricks do not provide a general improvement to PPO over standard methods, as we state in our paper, but we identify specific cases where they are valuable.
>
> To answer your question about other algorithms, we think it is likely that our results transfer to similar actor-critic algorithms such as A2C, in particular, due to the similarity between A2C and PPO [3]. Certain tricks like percentile scaling could be applied to other algorithms (Q-Learning, Vanilla Policy Gradients, etc), but it’s unclear how the tricks would affect them.
>
> If this changes your perspective on our work, please consider changing your score. Large empirical studies like this require publication to continue securing compute funding.
>
> [1] Engstrom, Logan, et al. "Implementation matters in deep policy gradients: A case study on ppo and trpo." arXiv preprint arXiv:2005.12729 (2020).
>
> [2] Andrychowicz, Marcin, et al. "What matters in on-policy reinforcement learning? a large-scale empirical study." arXiv preprint arXiv:2006.05990 (2020).
>
> [3] Huang, Shengyi, et al. "A2C is a special case of PPO." arXiv preprint arXiv:2205.09123 (2022).

---

> > ### Comment · Reviewer_DsH5 · 2023-08-19
> > **Rebuttal response**
> >
> > I appreciate your detailed response, and I acknowledge that PPO is a prominent algorithm that has been thoroughly investigated. However, I firmly believe that exploring the application of these tricks to a variety of algorithms is crucial. As it's common knowledge that the same tricks may yield disparate results when applied to different algorithms, limiting the study to a single algorithm might constrict the paper's overall contribution.
> >
> > I have also taken into consideration the perspectives of other reviewers and understand that the extensive experiments conducted in this paper can be seen as valuable to the research community. Nevertheless, I'm concerned that leveraging a significant amount of computational resources to execute an analysis that might not be highly innovative, and arriving at conclusions that might not be particularly surprising, does not seem fitting for acceptance at a prestigious conference like NIPS. Moreover, from a practical standpoint, the experimental results don't provide enough guidance for a user to choose anything other than the original PPO with reward clipping, neglecting the dreamerV3 tricks.
> >
> > Finally, it's important to note that some of my concerns were left unaddressed in the rebuttal, such as those related to the state-of-the-art performance. Thus, I must stand by my initial rating. I encourage the authors to consider expanding the scope of the paper to include more algorithms and strive for conclusions that can add more substantial value to the field.

---

> > > ### Author Response · Authors · 2023-08-20
> > >
> > > Thank you for your response, we appreciate you taking the time to read the other reviews and continue this discussion. While we would have liked to run these experiments with more algorithms, it was prohibitively expensive to do so, and we believe that our results on PPO are of significant interest to other RL researchers.
> > >
> > > Would you be able to elaborate on which results specifically you might have expected? We believe our paper contains many surprising results, particularly given the academic reception to Dreamer V3. We will repeat one of our comments to reviewer 75kB about why we and others were surprised by the results:
> > >
> > > > The five algorithmic tricks were presented by DreamerV3 as techniques to improve performance and decrease/eliminate the need for hyperparameter tuning. This sparked a tremendous amount of interest in the academic community. From their mathematical definitions, very few of the tricks appeared limited to world-modeling, and could seemingly be added to any Actor-Critic algorithm. Every other RL researcher we spoke to expected them to generalize broadly to other algorithms.
> > >
> > > We attempted to address your points regarding SOTA in our rebuttal by clarifying the intention of those experiments. Our goal was to use several environments to identify  cases where these tricks succeed or fail. We do not claim that these tricks provide SOTA results on either of these environment sets as a whole. Instead, our experiments demonstrate the types of environments where these tricks are useful. We will clarify this in the camera ready version of our paper. The paragraph that attempted to convey these thoughts is copied below:
> > >
> > >
> > > > The purpose of experimenting on Atari without reward clipping was to study a case with widely varying reward scales, which is where these tricks were expected to help most. We see in this case that symlog significantly improves PPO’s base performance here, as well as percentile scaling to a lesser degree. We see in our experiments with Atari and DMC that symlog is actually valuable in environments with large reward ranges, but less effective when rewards are already well normalized. These tricks do not provide a general improvement to PPO over standard methods, as we state in our paper, but we identify specific cases where they are valuable.
> > >
> > > Finally, our work does provide practical guidance to researchers interested in leveraging DreamerV3 tricks to improve performance. For example, based on our experiments, we recommend using symlog predictions in environments that have large or unbounded rewards, because symlog predictions perform comparable to or better than PPO in those environments. We hope this clarifies the practical impact and research interest of our work.

---

### Official Review · Reviewer_75kB · 2023-07-06

**Soundness:** 3 good
**Presentation:** 3 good
**Contribution:** 2 fair
**Rating:** 6
**Confidence:** 3

**Summary:**

The authors investigate the architectural and algorithmic improvements proposed in the DreamerV3 paper, but applied to PPO. The tricks are each compared on the DMC and Atari suites.

**Strengths:**

Thorough benchmark papers are a valuable asset to the RL community due to the difficulty in accurately comparing them and typically large error margins.

The paper is well-written and easy to read, and cites related work

DreamerV3 and PPO are both very relevant.

**Weaknesses:**

The conclusions of the experiments were not really surprising. While I think a benchmark / reproducibility paper like this still has value, I have some concerns with that:

* First, it misses the obvious opportunity to first reproduce the DreamerV3 tricks on the Dreamer RL algorithm itself. It would not have to be every benchmark, but getting similar results on e.g. Atari would have helped support the claim that these tricks were correctly implemented, as well as reproduced a recent method of interest.

* Second, why PPO? Is it even a good target for these tricks? According to the introduction, a big motivation for these were to enhance stability of actor-critic approaches, which typically struggle with stability. While PPO has some similarity to actor-critic approaches due to taking multiple steps and value function/GAE estimation, it was my impression that it behaved more like a policy gradient approach (which are typically more stable but on-policy and less efficient in theory). As such, the DreamerV3 tricks might have been a more natural fit for e.g., SAC.

---

After rebuttal: I think the authors gave adequate answers. Although they did not remedy all weaknesses, empirical studies like this are very important for RL research and it would be prohibitively expensive to test more algorithms. As such I think it would be useful for the community even though it has some limitations.

**Questions:**

During the course of writing this paper, did you try or manage to reproduce any results of DreamerV3?

Why the choice of PPO instead of e.g., SAC?

**Limitations:**

Seems clear.

---

> ### Author Rebuttal · Authors · 2023-08-09
>
> Thank you for your thorough review and suggestions. First, we would like to clarify a bit of nuance surrounding the paper. The five algorithmic tricks were presented by DreamerV3 as techniques to improve performance and decrease/eliminate the need for hyperparameter tuning. This sparked a tremendous amount of interest in the academic community. From their mathematical definitions, very few of the tricks appeared limited to world-modeling, and could seemingly be added to any Actor-Critic algorithm. Every other RL researcher we spoke to expected them to generalize broadly to other algorithms.
>
> In this work, we are specifically interested in transferring these tricks to other actor-critic methods, so we would also not consider it a reproduction of the DreamerV3 work. The authors open-sourced DreamerV3 shortly after publication, so we could have added PPO to their implementation in place of REINFORCE. However, it would have taken much longer to implement and been harder to verify correctness. Our code for this project is around 10x shorter than the DreamerV3 codebase and builds on CleanRL, one of the simplest and most widely used learning libraries. Our experiments consumed around 10,000 A100 hours and well over 100,000 CPU hours just to test on PPO. We had to only pick one algorithm, and we explain why we used PPO specifically in our overall response.
>
> As for the implementation quality, we believe that by cross-referencing our implementations with both the DreamerV3 paper and open-source code, individually testing each trick with regular regression checks against the original PPO algorithm, and open-sourcing our own code for community use, we have sufficiently safeguarded our implementations of these fairly simple tricks from mistakes.
>
> If this changes your perspective on our work, please consider changing your score. Publication makes it possible to continue securing funding for this type of large-scale empirical work.

---

> > ### Comment · Reviewer_75kB · 2023-08-13
> > **Thank you for the rebuttal.**
> >
> > I thank the authors for the reply. I do think benchmark papers important, even if I would have preferred to validate their implementation against some of the published results of Dreamerv3 first before extending the tricks to method that is quite different (model-free PPO). I will increase my score.

---

> > > ### Author Response · Authors · 2023-08-15
> > >
> > > Thank you for the comment and for increasing your score! Our implementation of the tricks was much more modular than DreamerV3, where the tricks were mostly intermixed with other implementation details. It would have required a fairly large refactor of their codebase to use the same implementations of the tricks in both our CleanRL code and the Jax-based DreamerV3 code. Nevertheless, we agree that if we had more time it would have been valuable to verify our implementations in this way, so thank you for the suggestion.

---

### Author Rebuttal · Authors · 2023-08-09

Thank you to all of the reviewers for their comments and suggestions about our work. We’d like to address our choice of PPO because several reviewers suggested other potential starting algorithms for this work (DreamerV3, A2C, SAC).

To start, this work is not a reproduction of DreamerV3 but instead an attempt to transfer them to PPO and a careful study of their interactions. The tricks introduced by DreamerV3 were designed to stabilize actor-critic learning so we chose PPO, the most popular actor-critic algorithm. A2C is another viable choice, but it is much less popular than PPO and can be implemented as a special case of PPO [1], so PPO was the clear choice between the two.

SAC is another viable option. DreamerV3 borrows some ideas from SAC, such as its entropy regularized objective. However, SAC trains two Q functions in place of a typical V function critic, which makes the implementation of many tricks less straightforward. SAC and DreamerV3 also require a forward and backward pass for each environment step, causing them to train slower (in terms of wall-clock time) than other actor-critic algorithms.

PPO is the most popular RL algorithm because of its simplicity and efficacy in a broad range of environments. Unlike other algorithms, it has been used at scale in Dota2, silicon chip design, and ChatGPT. PPO is an improvement to REINFORCE, the policy gradient algorithm that DreamerV3 uses for policy updates, and uses the same standard V function critic as DreamerV3. PPO also trains faster than DreamerV3 and SAC. Ultimately we felt that transferring the tricks to PPO would require the least modifications to the tricks introduced by DreamerV3 and have the highest impact.

[1] Huang, Shengyi, et al. "A2C is a special case of PPO." arXiv preprint arXiv:2205.09123 (2022).

---

### Decision · Program_Chairs · 2023-09-21

**Decision:**

Accept (poster)

**Comment:**

A detailed empirical investigation on a relevant algorithm (PPO with DreamerV3 tricks), executed decently, should be broadly useful to the community even if the results are not surprising. Its main caveat is not reproducing the actual DreamerV3 baseline within the same setting, hence leaving some reviewers with residual doubts even after the rebuttal.